coastal erosion; coastal resilience; coastal hazards; LiDAR; shoreline change

**Corresponding author:**
Thomas Oliver;
Email: t.oliver@unsw.edu.au

# Extreme storm impact and recovery on a natural beach-foredune system: The June 2016 storm at Bengello Beach, southeastern Australia

Thomas Oliver[1] ⓘ, Michael A. Kinsela[2] ⓘ, Thomas B. Doyle[3,4] ⓘ, Dylan McLaughlin[4] ⓘ and Roger F. McLean[1]

[1]School of Science, University of New South Wales, Canberra, Australia; [2]School of Environmental and Life Sciences, University of Newcastle, Callaghan, NSW, Australia; [3]Water, Wetlands and Coastal Science; Science and Insights Division; Department of Climate Change, Energy, the Environment and Water, NSW Government, Sydney, NSW, Australia and [4]Faculty of Science, Medicine and Health, University of Wollongong Australia, Wollongong, Australia

## Abstract

The June 2016 extratropical cyclone with anomalous ENE storm wave direction caused widespread beach-foredune erosion in southeastern Australia. At Bengello Beach, erosion volumes were 97–108 $m^3$/m for the central and southern parts of the beach, while the northern end only lost 18 $m^3$/m of sand. In the central and southern parts of the embayment, a surf zone bar formed 50–100 m further seaward than is typical for this beach and was a temporary store of sand eroded from the beach-foredune. A nearshore wave model showed substantial variability in wave power along the 10 m depth contour and explained the partial sheltering of the northern end of the embayment from storm impact. An embayment-wide time-series of airborne LiDAR further emphasised the alongshore variability in beach-foredune erosion. The wide beach and broad, double-crested, well-vegetated foredune along much of the embayment was pivotal in protecting the shoreline. In the centre and south of the beach, recovery took nearly three years and although complete by volume, the foredune was narrower and less resilient. The results emphasise the role of wide beaches and natural vegetated foredunes in buffering extreme storms and suggest foredune rehabilitation should be a key management priority for sustainable coasts.

## Impact statement

Extreme storms cause erosion of sandy coastlines around the world. The June 2016 storm was one such event impacting southeastern Australia and had an unusual ENE wave direction leading to a different spatial impact on beaches of the region. This was apparent at Bengello Beach, a site of long-term beach-foredune monitoring, with the centre and southern parts of the beach loosing 100 $m^3$ of sand per metre of beach, while the northern end only lost 18 $m^3$/m. Variability in modelled wave power along the 10 m depth contour during the event explains this spatial variation in erosion impact. A wide beach and broad, high, well-vegetated foredune was a critical buffer absorbing the storm erosion. Beach and foredune recovery took nearly three years, but the foredune was narrower than before the event and less resilient to future events. These results emphasise the important role that wide beaches and broad, high, well-vegetated foredunes can play in moderating extreme storm impact. Management actions on natural sandy coasts should prioritise maintaining beach width and foredune stability to help protect coasts into the future.

## Introduction

Intense cyclones and storms, both tropical and extratropical, cause significant erosion on sandy coastlines worldwide. These impacts include removal of beach sand (swash regime), foredune cut leading to a vertical scarp or sandy cliff (collision regime), foredune overtopping (overwash regime), and breaching or even destruction of the foredune system (inundation regime) (Masselink and van Heteren 2013; Castelle & Harley 2020; Leaman et al. 2021; Castelle & Masselink 2023; Hesp 2002, 2024; Davidson-Arnott et al. 2024; Turner et al. 2024). The controls on foredune erosion have been reviewed more broadly with water level height and duration of high water seen as critical as this determines what elevation wave action can impact or erode the beach and adjacent dunes (Davidson et al. 2020). It was also noted that antecedent dune toe elevation, and berm volume play key roles in controlling the severity and alongshore variation in storm erosion (Beuzen et al. 2019; Khan et al. 2025). The role of vegetation in mitigating foredune erosion has also been recently debated (Moore et al. 2025). Dune vegetation is widely believed to reduce foredune erosion during storms (Feagin et al., 2015, 2019; Davidson et al. 2020; Figlus

et al., 2022), with studies showing that greater plant size (Kobayashi et al., 2013), density (Silva et al., 2016; Charbonneau et al., 2017), and diversity (Maximiliano-Cordova et al., 2019) are generally associated with lower erosion rates. However, a recent experiment (within a wave flume) has shown that during extreme storm conditions, dune vegetation accelerated foredune erosion (Feagin et al., 2023).

Systematic monitoring from satellites (Bishop-Taylor et al. 2021; Wulder et al. 2022; Vos et al. 2023) and episodic aerial photography (Moore et al. 2006; Hanslow 2007) enables assessment of interannual to decadal scale trends in shoreline position, yet there remains an ongoing need for detailed localised field-based observations of event-scale erosion impact and recovery (Short, 2022), especially in natural beach-foredune settings (Woodroffe et al., 2022). Existing field-based long-term monitoring programs are sparse, both globally and regionally, and yet have still provided critical data on storm impacts and recovery in the context of decadal trends (e.g. Larson & Kraus 1994; Kroon et al. 2008; Quartel et al. 2008; Hesp 2013; Ollerhead et al. 2013; Banno et al. 2020; Eichentopf et al. 2020; Zhang & Larson 2021; Bertin et al. 2022; Suanez et al. 2023; McCarroll et al. 2023; Davidson-Arnott et al., 2024). For example, extreme storms in the 2013/2014 winter season around the Atlantic coast of Europe caused widespread coastal erosion (Masselink et al. 2016). The recovery phase has been documented for the Truc Vert coast in SW France (Castelle et al., 2017) and along the SW England coast, where the role of higher-energy events appeared critical in mobilising deeper water storm deposited bars, but recovery processes were nonetheless stochastic and complex (Scott et al. 2016). The role of extreme storms in mobilising deeper stores of sediment to nourish beaches has also been emphasised by Harley et al. (2022) with data from Australia (Narrabeen-Collaroy), the UK (Perranporth) and Mexico (La Mision), reflecting established physical concepts of wave-driven sediment transport in the nearshore-shoreface zone (e.g. Wright & Short 1984). Better understanding of these nearshore-beach-foredune interactions is critical for both emergency and long-term coastal management.

To that end, three-dimensional (3D) surveys with piloted or remotely piloted aircraft (RPA, or drones) carrying a high-resolution camera or LiDAR (or Light Detection and Ranging) sensors permit coastal change to be documented in a range of environments and across larger spatial areas and more regular temporal scales (Turner et al., 2016; Doyle et al., 2019; Joyce et al., 2023; Asbridge et al., 2024). Such approaches deliver rapid, centimetre-accurate, 3D elevation models that can be repeated after storms or at regular intervals to quantify beach-foredune erosion, dune evolution or regional foredune spatial patterns and nearshore bathymetry/ habitat mapping (Harley et al., 2017; Linklater et al., 2019; Doyle et al., 2019; 2024; Kinsela et al., 2022). RPAs allow access to hard-to-reach sites and allow for rapid deployment pre and post storms (Downes et al., 2025), while crewed aircraft can cover hundreds of kilometres in single missions (Harley et al., 2017). The resulting high-resolution datasets are beginning to fill critical knowledge gaps in coastal planning and hazard assessment and facilitate analysis of interannual beach-foredune changes and also near-real-time monitoring of storm response and recovery (McLaughlin et al., 2025a,b; Khan et al., 2025).

This study aims to improve our understanding of the impacts to and recovery of a sandy, natural beach and foredune system following a single severe storm. The June 2016 storm event impacted the entire New South Wales (NSW) coast causing widespread beach and dune erosion and damage to foreshore coastal property and infrastructure (Harley et al., 2017). We utilise field topographic survey data from the long-running (1972-ongoing) beach monitoring program at Bengello Beach, NSW and deep-water to near-shore wave modelling to extend the analysis of Harley et al. (2017) by characterising the storm impact in a natural beach setting in southern NSW. Insights from beach profiling and ground photography data are supplemented by analysis of satellite imagery showing changes in surf zone and accompanying bathymetric survey data and repeat airborne LiDAR surveys. The June 2016 storm was notable for this coastline and this paper contextualises its impact through comparison to other storm events recorded in the long-term monitoring record, including recent events in 2022 (Oliver et al. 2024), and considers the role of antecedent beach topography and foredune stability in mitigating erosion from extreme storms.

## Regional setting

### Study site

Bengello Beach is located on the south coast of NSW approximately 250 km south of Sydney. The gently crescent-shaped shoreline is 6 km long and faces ESE. The beach is bounded at the southern end by the training wall of the Moruya River and by Broulee Head in the north. Subaqueous rocky reef is present adjacent to Broulee Head and Broulee Island in the north, and Moruya Heads in the south, while the shoreface of Bengello Beach itself is sediment-dominated and is comparatively steep (Figure 1) (Roy et al. 1994; Oliver et al. 2020). The surf zone is typically in a Transverse Bar and Rip or Rhythmic Bar and Beach state (Wright and Short 1984) which varies temporally and alongshore with the northern end generally slightly higher energy and finer grained and hence more dissipative (Wright et al. 1979). The outer surf zone bar is ~75–150 m from the shoreline. The beach-foredune sediment is dominated by fine to medium mature quartz grains with a small proportion of contemporary carbonate (<10%). A beach berm and cusps are common features with the berm crest reaching 2–2.9 m AHD and ~ 20–40 m wide (McLean & Shen 2006). This coastline experiences a mixed semi-diurnal tidal regime with spring and neap tide ranges of 1.6 m and 0.7 m, respectively.

Inland of Bengello Beach is a ~ 2 km wide sequence of foredune ridges deposited over the mid- to late-Holocene derived from the inner shelf and shoreface (Thom & Roy 1985; Thom et al. 1981; Oliver et al. 2015; 2020). The most recent foredune developed in the recovery phase following the 1974–1978 storms which impacted Bengello Beach (then known as Moruya) and continued to grow vertically and stabilise in the subsequent decades (McLean and Shen 2006; McLean et al. 2023). The foredune vegetation follows a typical succession from pioneering species such as *Spinifex sericeus (Spinifex)*, *Cakile maritima* (sea rocket), *Cakile edentula* and *Carpobrotus glaucescens* (coastal pigface) occupying the foredune's seaward edge, crest and the incipient foredune (when present) (Figure 2). Transitioning landward from the foredune crest, *Lepidosperma gladiatum (*coastal sword sedge)*, *Lomandera longifolia* (mat rush) and *Acacia longifolia subsp. sophorae (*coastal wattle) appear with juvenile *Banksia integrifolia (Banksia)* present on the lee slope of the foredune and mature *Banksia* in the swale (Figure 2) (Doyle and Woodroffe, 2025). Four profiles have been monitored at Bengello Beach since January 1972 on a monthly to bi-monthly basis (McLean et al. 2023; Thom & Hall 1991). These are located near the centre of the beach and are denoted P1 to P4 (Figure 1). P1 is the southernmost and is separated alongshore from P2 by 286 m. P2, P3 and P4 are approximately ~70 m apart (McLean et al., 2023).

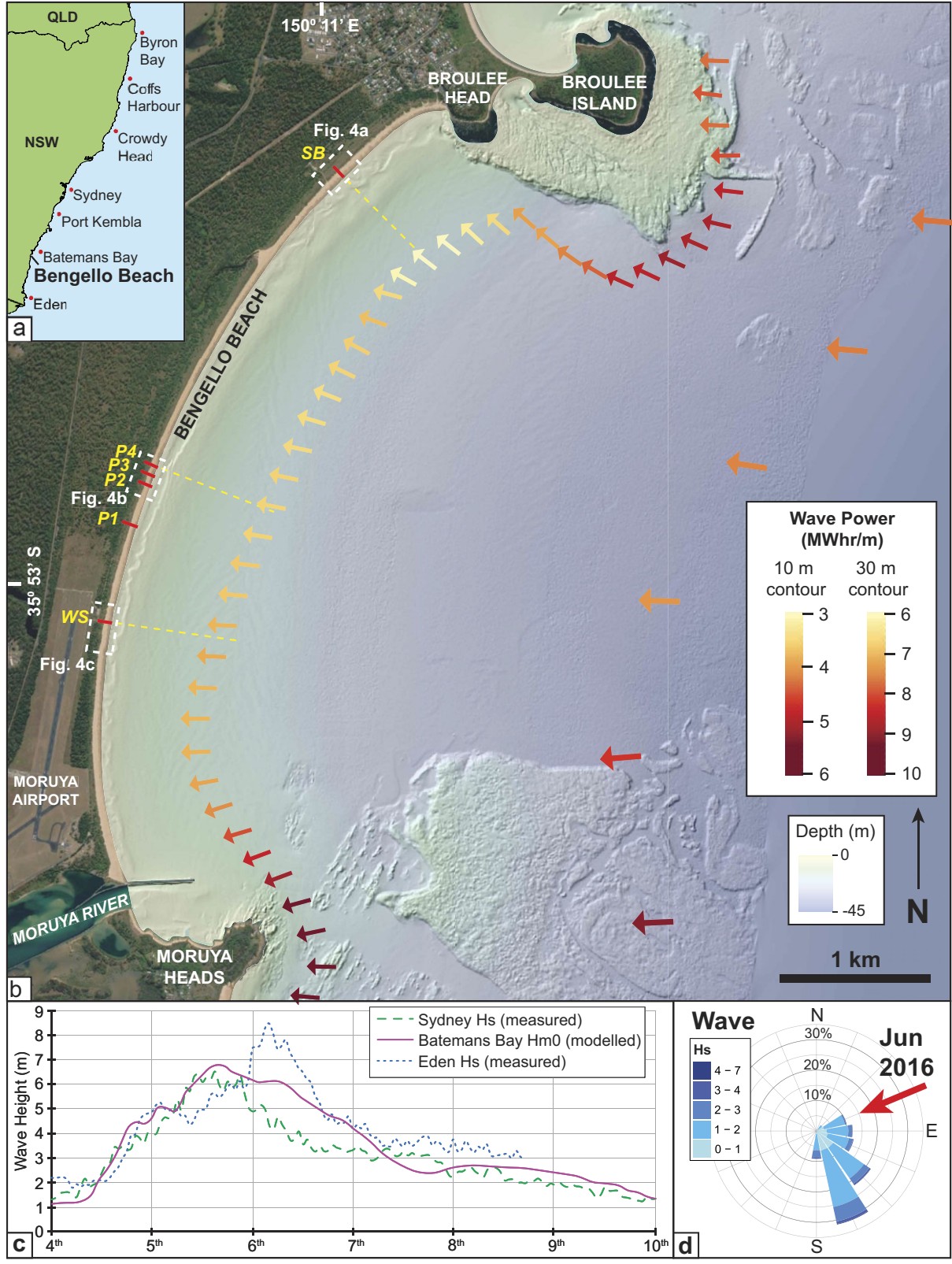

**Figure 1. (a)** Location of Bengello Beach and the approximate position (red dots) of the 7 offshore wave buoys along the NSW coast. **(b)** Map of the Bengello Beach embayment showing the location of the 6 survey profiles (red lines) where beach surveys have been measured. P1-P4 are the long-term profile locations measured since 1972, while South Broulee (SB) and Windsock (WS) have been measured since mid-2012. Bathymetric data is from Marine LiDAR collected in 2018. Location of three bathymetric profiles (yellow dashed lines) corresponding to the SB, P4 and WS beach-dune profiles are indicated and appear in Figure 3f. Coloured arrows along the 10 m and 30 m depth contours spaced at 200 m and 1,000 m respectively represent the total storm wave energy flux in MWhr/m (10 m: $P$; 30 m $P_d$) and weighted average peak direction. **(c)** Time series of offshore significant wave height from regional wave buoys (Sydney, Eden) and wave height at the Batemans Bay buoy for the June 2016 storm event. **(d)** Wave rose from the Batemans Bay deep water wave buoy for the period 2001–2019 with the wave direction during modelled peak wave height for the June 2016 event at Batemans Bay indicated by the red arrow.

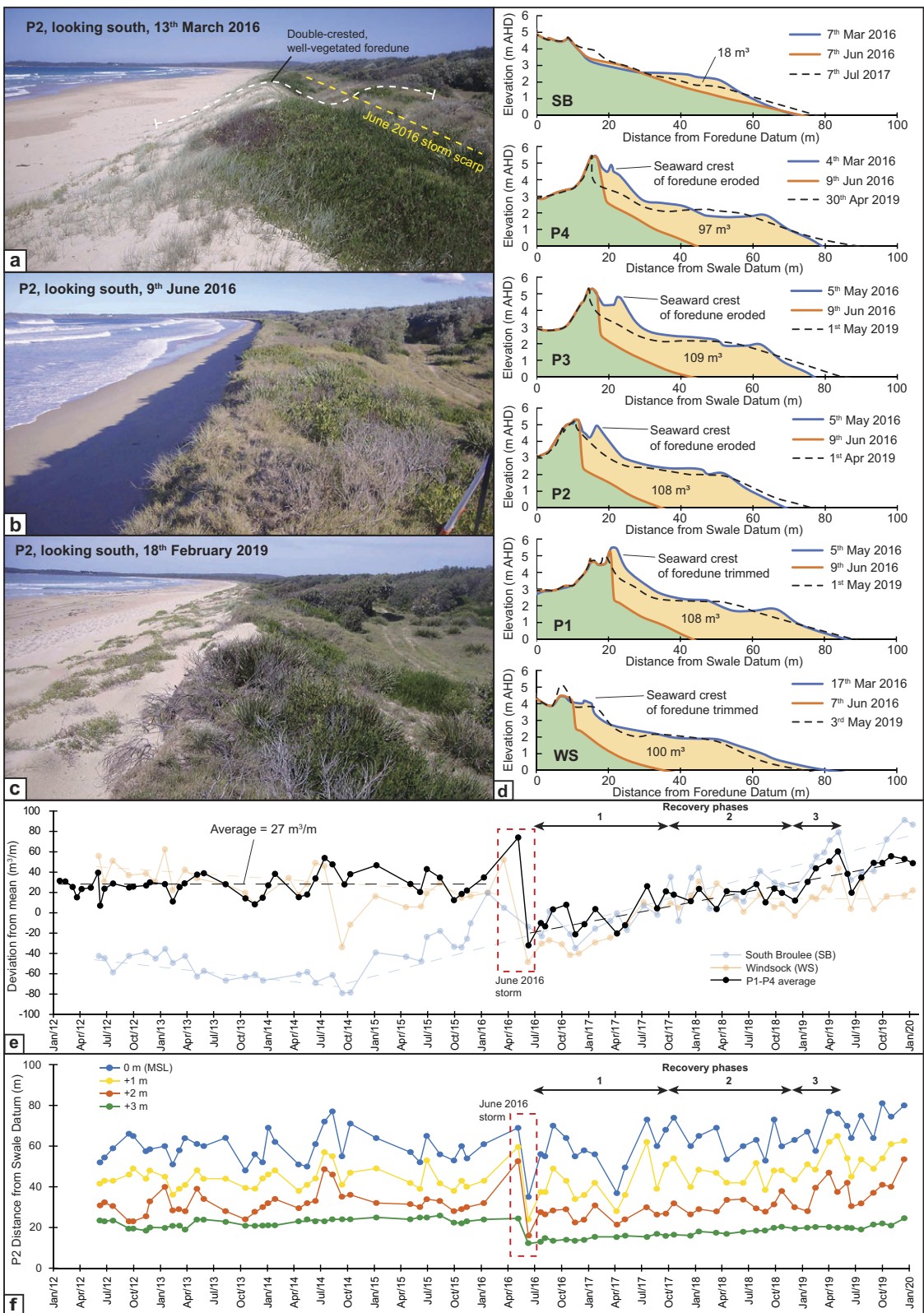

**Figure 2.** Photos of Bengello Beach from Profile 2 looking south **(a)** before the June 2016 storm, **(b)** immediately after the event, and, **(c)** once the beach/dune system had almost fully recovered. Of particular note is the double-crested foredune present in the pre-storm photo from P2, which did not re-form despite a return to the approximately the pre-storm volume. **(d)** The impact of the 2016 storm is characterised for the six survey profiles (see Figure 1 for locations) and are arranged north to south. The pre-storm survey is shown in blue, the post-storm survey in orange, and the material eroded shaded in light orange. The scarp which developed following the June 2016 storm was 2.4–2.8 m at the four central profiles (P1-P4) and WS. The dashed black line represents the beach-foredune profile when the beach/foredune returned closest to the pre-storm volume and, with the exception of SB, occurred in May or April of 2019. **(e)** A timeseries of beach/foredune volume relative to each profiles mean volume from 2012 to Jan 2020 for SB, the average for P1-P4 and WS. The June 2016 storm is highlighted in a red dashed box. **(f)** A timeseries of distance from the SD to the 0 m (MSL), +1 m, +2 m, and + 3 m intercepts from topographic surveys from P2. The 2016 storm is highlighted in a red dashed box with three recovery phases denoted with black arrows.

The deep-water wave climate along the NSW coast is captured by a series of wave buoys with the Batemans Bay buoy closest to the study site. Buoy data show the dominance of SSE wave directions with an average significant wave height (Hs) of 1.5 m and peak wave period of 9.5 s (Figure 1d) (Lord and Kulmar 2000). Most storm waves (>90%) arrive from similar SSE directions and on average, 15 storm events (Hs > 3 m) are recorded by the buoy each year (Shand et al. 2010). A network of Sofar Spotter wave buoys have been deployed in nearshore locations in recent years, including one adjacent to Bengello Beach in ~13 m water depth since November 2020 with results collected from these deployments providing critical training and validation data for nearshore wave modelling (Kinsela et al. 2024). Data comparison between the offshore and nearshore buoys capture slight-moderate wave height attenuation and more easterly wave directions as waves transform and refract towards the coast (Kinsela et al. 2024; Oliver et al. 2024).

### The June 2016 storm

The June 2016 storm event occurred during 4–7 June and impacted the entire coast of NSW. It was unusual in terms of its synoptic pattern and resulting wave direction, with storm waves arriving from ENE (57–93°) directions rather than from ESE-SSE directions as is typical for this coast (Harley et al., 2017; Mortlock et al. 2017). Peak Hs recorded by offshore wave buoys reached 5 m in the far north at Byron Bay, 6–7 m along most of the coast and over 8 m at Eden (Figure 1a) (Louis et al. 2016; Mortlock et al. 2017). The large easterly waves were generated by unusual fetch created by the extra-tropical cyclone and adjacent broad anti-cyclonic intensification. The coincidence of large tides with a moderate storm surge of ~0.5 m recorded nearby at the Batemans Bay tide gauge (Louis et al. 2016), with large easterly waves caused extensive damage along the NSW coast, particularly in the highly populated central region (Harley et al. 2017).While the wave and water level statistics in isolation were not extremely rare, the combined conditions and in particular easterly wave direction exposed partially sheltered open coast beaches to severe coastal erosion (Burston et al., 2016).

Using pre- and post-storm LiDAR datasets, Harley et al., (2017) analysed regional-scale beach response along the central to northern NSW coastline, finding significant spatial variability in erosion impact resulting from alongshore gradients in storm wave energy flux due to variable coastline alignment and the anomalous wave direction. At the long-term monitoring profiles at Narrabeen-Collaroy Beach, the event caused the largest single beach erosion (mean = 121 m$^3$/m) in the survey record (1976-ongoing) (Harley et al. 2022). Along the southern NSW coast, observations at the coast and offshore were sparser than the central to northern coasts. However, the offshore wave buoy at Eden (150 km south of Bengello Beach) recorded the highest peak Hs along the coast at 8.5 m, and the largest individual wave height (17.7 m) ever recorded up to that time in NSW (Figure 1). The peak residual storm surge water level approaching 0.5 m at Batemans Bay (Figure 1) was also the highest recorded along the NSW coast. Together this suggests that storm conditions during the event were more intense along the southern coast compared to the central to northern coasts.

## Methodology

### Storm wave modelling

Deep-water wave conditions along the NSW coastline during the June 2016 storm were captured by 6 of the 7 permanent offshore wave buoys (see locations in Figure 1a). Unfortunately, data capture was poorest in the vicinity of the study site, with the Port Kembla buoy suffering data loss and the proximal Batemans Bay buoy being offline due to prior technical issues. However, data capture was good between Byron Bay and Sydney (see Harley et al., 2017, Figure 1), and at the Eden buoy located 150 km south of the study site (Figure 1a).

To investigate deep-water wave conditions in the Batemans Bay region and variation in nearshore wave energy and direction along Bengello Beach during the storm, the storm wave conditions were modelled using WAVEWATCH III® (WW3; Tolman 2014) forced with hourly gridded wind data from the NOAA-NCEP Climate Forecast System v2 (CFSv2; Saha et al. 2014). The model comprises three nested grid domains – global (1°), Australia (0.25°) and south-east Australian (0.05°) – with an unstructured mesh in coastal waters reaching 100 m resolution at the 10 m bathymetry contour, and uses the ST2 source terms package (Baird Australia, 2017). The model was found to achieve a root-mean-square error of 0.55 m for significant wave height (Hm0) when compared to measured wave data from the Coffs Harbour, Crowdy Head and Sydney offshore wave buoys during the June 2016 storm (Harley et al., 2017).

The gross wave energy flux ($P$) and wave energy flux directed towards the coast ($P_\alpha$) were calculated for each hour of the storm along the 30 m and 10 m depth contours, at 1 km and 200 m alongshore spacing respectively, and integrated across the storm duration to yield total storm wave energy flux values (Figure 1b) following Splinter et al (2014) and Harley et al. (2017). The approach combines the magnitude of wave energy fluxes and event duration for comparing total exposure to the June 2016 storm wave energy along Bengello Beach. A threshold of H$_{m0}$ = 3 m was used to define the storm duration. The weighted mean of peak wave directions ($\theta_p$) at each model output node was calculated by weighting the modelled directions by the hourly gross wave energy flux ($P$).

### Sentinel-2 data

Sentinel-2 images were extracted from Digital Earth Australia Baseline Satellite Data collection and displayed as a simple RGB image. All available cloud-free images were examined, and a selection was arranged to show beach and surf zone conditions prior to June 2016, soon after the event, and then subsequently until around one year later (Figure 3).

### Bathymetric surveys

Two jet skis mounted with single beam echosounders were used to survey the nearshore bathymetry at Bengello between 18$^{th}$ – 20$^{th}$ November 2014 by the NSW State Government (Coast and Marine Science Team). During the same period a quad bike survey captured the beach topography from ~0 m AHD up to ~4 m AHD. Survey data was gridded from point data to produce an elevation surface with a resolution of 12 m × 12 m appropriate to the survey resolution. Recorded in GDA 94 MGA Zone 56, sounding datum AHD. Marine LiDAR data collected in 2018 for the NSW State Government was downloaded from the "Elvis - Elevation and Depth - Foundation Spatial Data" portal as a 5 m × 5 m grid. See below for details of the collection and processing of this data. Identically positioned topographic profiles were drawn through both datasets for comparison of nearshore bathymetry corresponding to the South Broulee (SB) profile, Profile 4 (P4) and the Windsock (WS) profile (Figure 3f, see locations in Figure 1).

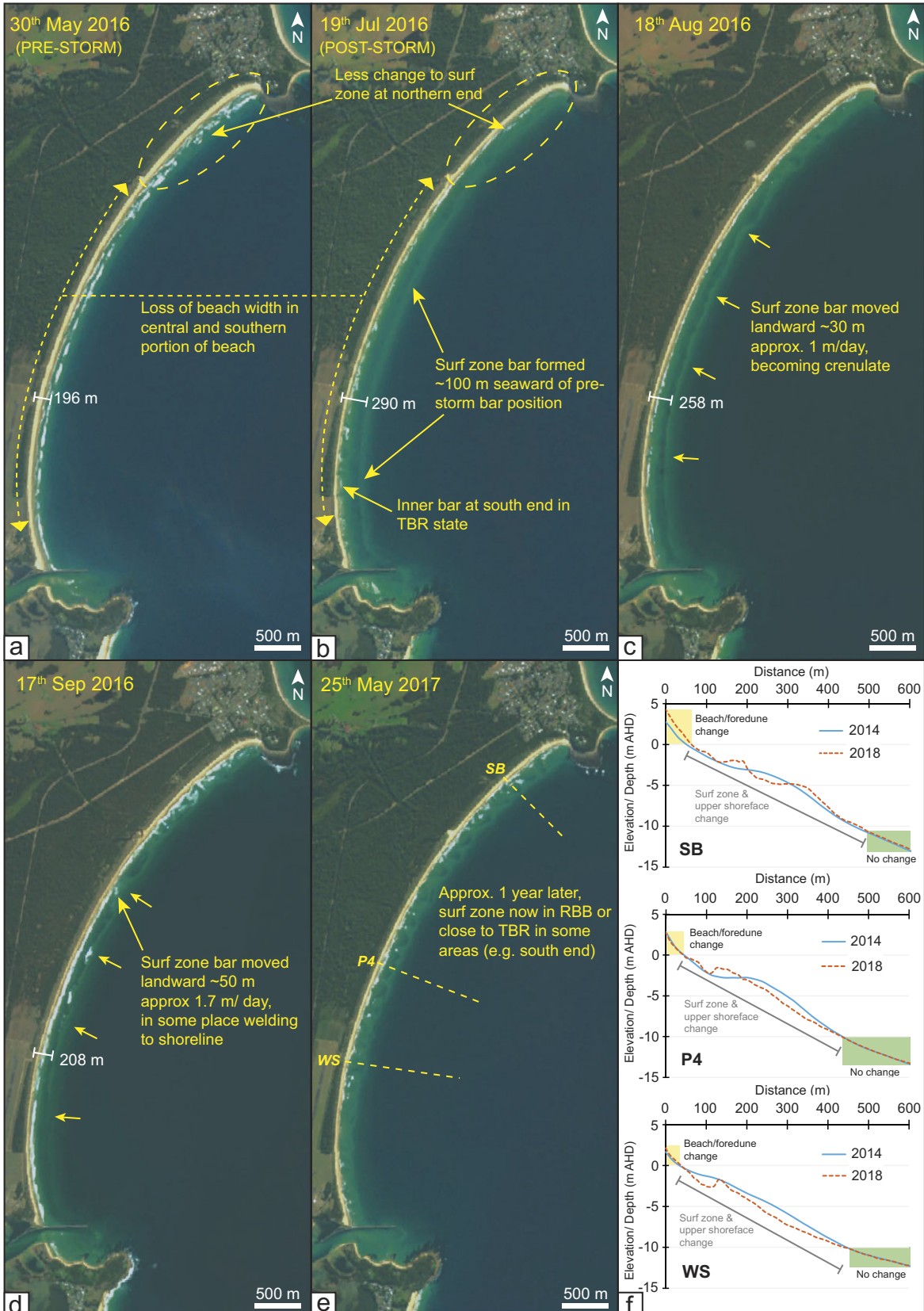

**Figure 3.** Selected Sentinel-2 images covering the period before and after the June 2016 storm event **(a-e)** and nearshore profiles **(f)**. The Sentinel-2 images demonstrate changes in the arrangement of surf zone bars caused by the June 2016 storm with the white line and value in meters representing the distance from a fixed position to the landward edge of the surf zone bar. The bathymetric profiles (locations shown in e) and also on Figure 1(b) are derived from two data sources. The 2014 data is a single-beam beach boat and jet ski survey, and the 2018 data is from marine LiDAR and is displayed in Figure 1.

### LiDAR collection, processing, interpretation

Airborne LiDAR used in this study to characterise NSW beach-foredune morphology comprised three complementary datasets (Figure 4). Firstly topographic LiDAR captured in 2011 (12th July) under the NSW Government "Coastal Capture" program (using a ALS50-II; Leica Geosystems sensor) was flown at ~1960 m a.s.l. (swath ≈ 942 m, 15 m overlap), and delivered point clouds with minimum point density of 1 point per $m^2$, and verified vertical and horizontal accuracies of 0.3 m and 0.8 m, respectively (Doyle and Woodroffe, 2018). Secondly, repeat surveys conducted by UNSW aviation in 2016, 2017, 2019 and 2020 using a Riegl Q480i lidar sensor integrated with a NovAtel SPAN AG62 GNSS/IMU, flown in a Piper PA-44 at ~300 m altitude, yielding ~1 point per 1.6 $m^2$ and achieving better than 0.2 m vertical and 0.5 m horizontal accuracy (Middleton et al., 2013). Note the 2016 LiDAR data from UNSW aviation was flown just after the storm event thus capturing its impact. Thirdly, the Marine LiDAR dataset referred to above is a seamless topographic-bathymetric survey (on the 5th September 2018) and was acquired by Fugro Australia Pty. Ltd. under contract to the NSW Government, using a combined Riegl VQ-820-G (land) and LADS HD-ALB (sea floor) sensors, flown at 1600–1800 ft. and 160 kn, with 336 m line spacing; and yielded minimum point densities of ≥2 points per $m^{-2}$ (Riegl), and yielded minimum vertical and horizontal accuracies of 0.36 m and 0.88 m, respectively (Kinsela et al., 2022; Fugro, 2019). All datasets were referenced to Geocentric Datum of Australia (GDA) 1994 (MGA zone 56), reduced to the Australian Height Datum (AHD) and quality-assured (i.e. ICSM classification level 3 for 2011 Lidar, or QA4LIDAR for 2018 Lidar, see FrontierSI, 2019), providing a consistent, high-precision foundation for successive coastal morphodynamic analyses.

Raw LiDAR files for years 2016, 2017, 2019, 2020, were classified into 'bare ground', using customised macro tools and settings in Terrasolid (as per methods for NSW government state LiDAR), which include a maximum building size of 200 m; terrain angle of 88 degrees, an iteration angle of 6 degrees to the plane, and an iteration distance of 1.4 m, to the plane. Each of these datasets were then manually cross-checked with state LiDAR, which have been classified to the ICSM classification level 3, meaning 99% of ground points have been cross-checked, (ICSM, 2011). This was completed using the profile viewer tool in Global mapper (v23) and verified by co-authors to ensure the results reflects real world topographic surfaces (i.e. bare ground).

Changes in foredune morphology were analysed using Digital Elevation Models (DEMs) produced from LiDAR derived point clouds (collected in 2011, 2016, 2017, 2018, 2019, 2020, and 2024). Ground-classified points were used to create triangulated irregular network (TIN) surfaces for the foredune area of interest, bounded on the seaward side by the 2 m (AHD) contour (from the 2011 dataset) and landward by the leeward swale of the active foredune. Raster DEMs were generated from TIN surfaces and resampled to a uniform 0.25 m resolution (based on the lowest resolution dataset – 2011) to ensure consistency across time steps. DEMs of Difference (DoDs) were calculated by subtracting the earlier DEM from the later one (e.g., 2016–2011) to quantify elevation changes.

### Beach surveys

Regular beach-foredune surveys during the period from ~2012 to the beginning of 2022 were collected using a rotating laser level for elevation and a fixed-length pole or tape measure for distance. These were conducted at the four long-term survey locations (P1-P4) near the centre of the beach where monitoring commenced in January 1972 (McLean et al. 2023), and two additional locations, one in the north of the embayment called South Broulee (SB), and another in the southern part of the embayment called Windsock (WS), both commenced in June 2012. A series of fixed benchmarks positioned at each profile provide a consistent elevation (in AHD) and distance control for each survey. At P1-P4 these benchmarks or 'datums' are denoted 'Back datum' (BD), 'Swale Datum' (SD), and 'Foredune Datum' (FD) according to their position in the landscape (see McLean et al. (2023) for a discussion of the datums and their history). During the period considered in this study (2012–2020), surveys at P1-P4 were measured from the SD and FD where active topographic change was occurring. Beach-foredune volume is calculated by determining the area in $m^2$ beneath each topographic profile bounded by 0 m AHD (which approximates mean sea level (MSL)) and a vertical line extending down to 0 m AHD from the datum. Following convention, this is converted to $m^3$ by assuming a 1 m wide profile. Timeseries of change in beach-foredune volume in this study has been presented as deviation from the mean value of each profile (Figure 2e). Distance from the SD or FD at each profile to the MSL position, +1 m, +2 m and + 3 m positions, termed here intercepts, were also calculated for each profile.

## Results

### Pre-storm beach-foredune conditions

At Bengello, the pre-storm beach was in an accreted state. Beach-foredune volumes for the period from 2012–2016 were relatively stable for P1-P4 averaging 27 ± 11 $m^3$/m above the long-term average (Figure 2e). Field surveys from May 2016 just prior to the event showed the average volume for profiles P1-P4 was 74 $m^3$/m above the long-term average (Figure 2e) which was the highest ever beach volume in the survey record (McLean et al. 2023). Similarly, the MSL intercept was 11 m further seaward compared to the long-term average (McLean et al. 2023). Prior to the storm, the foredune along the central segment of the Bengello compartment comprised two distinct crests spaced ~6.5 m apart and separated by a shallow swale (Figure 2d; Figure 4b). Both crests of the foredune were sporadically covered with low shrubs, including *coastal wattle*, pigface and mat rush, with long runners of *Spinifex* extending down through mounds of sea rocket to the backshore (Figure 2a). The two distinct foredune crests represented very different aged features. The inner crest was initially established in the early 1980s during the recovery after the major storms on the NSW coast from 1974–1978 (McLean et al. 2023) and persisted unscathed until 2022 where it was removed at P3 and P4 (Oliver et al 2024). On the other hand, the seaward crest of the foredune was a much more recent feature having built up from an incipient dune after storms in 2010 and 2012 to reach its maximum elevation and extent immediately prior to the June 2016 event. For other areas of the Bengello embayment, the southern segment, for example, also developed a double crested foredune prior to the 2016 event (Figure 2d, e; 4c), but the front crest seemed to have been re-built since a storm in 2014 (especially WS profile in Figure 2e). While the northern part of the embayment, despite also being impacted by the 2014 storm event, was also in a very accreted state prior to the 2016 event, as Figure 4a demonstrates with a wide beach and berm with rhythmic beach cusps, and early stages of an incipient foredune fronting the established dune (esp. SB profile, Figure 2e).

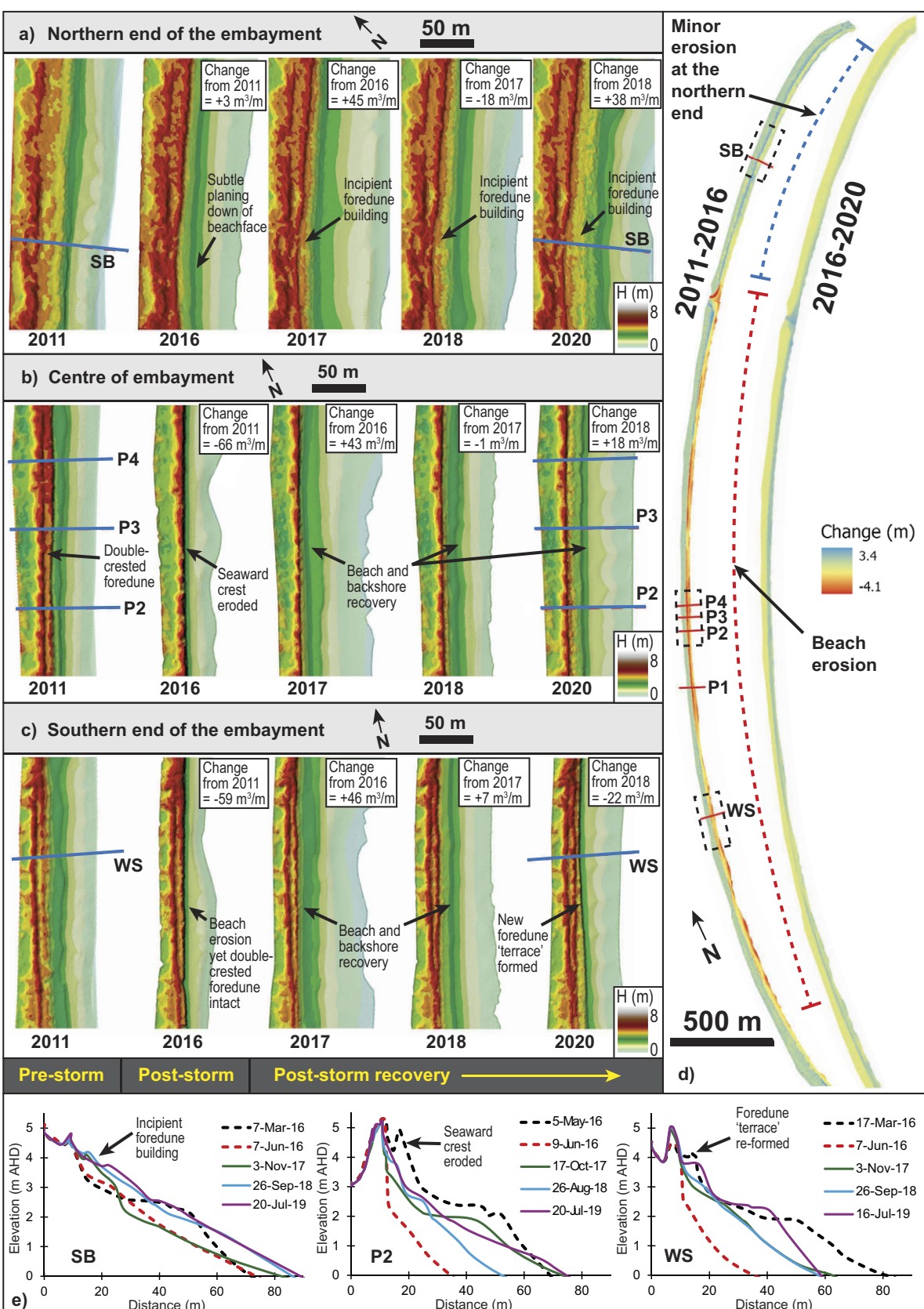

**Figure 4. (a-c)** ~ 200 m sections of DEMs derived from airborne LiDAR showing the foredune and beach at Bengello (see Figure 1 for locations). Volume change within each segment from year-to-year is indicated in m³/m, **(d)** Elevation difference comparing the 2011 and 2016 DEMs and 2016 and 2020 DEMs with yellow-red shading indicating volume loss and green-blue shading indicating volume gain. The location of the six beach profiles is indicated and black dashed boxes show demark the DEM sections from **(a-c)**. **(e)** Beach/foredune profiles from field surveying showing the pre- and post-storm topography (same as Figure 2), with the addition of yearly profiles which correspond closely to the LiDAR DEM collection dates and provide a complimentary cross-sectional view of the beach/foredune recovery phase.

### Storm wave conditions

Modelled deep-water wave data at the Batemans Bay offshore wave buoy location are shown in Figure 1, along with measured wave data from the Sydney and Eden offshore wave buoys, located 250 km north and 150 km south of the study site respectively. Measured data show the changing shape of the development and decay of storm wave conditions between Sydney and Eden as the system migrated southwards off the NSW coastline. The modelled data at Batemans Bay depicts an intermediate storm wave shape between those observed at Sydney and Eden, with deep-water $H_{m0}$ approaching 7 m at the storm peak and storm conditions ($H_{m0} > 3$ m) in 30 m water depth off Bengello Beach persisting for 60 h. The comparison provides confidence that the model captured key characteristics of storm wave conditions that were experienced at Batemans Bay during the June 2016 event.

The modelled peak storm $H_{m0}$ off Bengello Beach was 5.7–6 m, where the total wave energy flux over the duration of the storm averaged 9.08 MWhr/m (8.64–9.96 range) along the 30 m depth contour, with weighted mean peak directions capturing the easterly arrival direction of incident wave energy (Figure 1). While the wave energy entering the embayment was relatively consistent alongshore, the east-facing orientation of the bathymetry and shoreline south of WS resulted in slightly higher $P_\alpha$ values there. Along the 10 m contour however, an alongshore gradient in total storm wave energy flux ($P$) was observed (Figure 1), being lowest in the north (SB), moderate along the central sector (P1-P4) and highest from WS south. Weighted mean peak wave directions were relatively normal to the beach shoreline following refraction to the 10 m depth contour. The reduced wave energy along the northern beach shoreline occurred adjacent to a zone of wave energy convergence around Broulee Island (also observed at Moruya Heads) where the total storm wave power was twice that at the beach shoreline.

### Surf zone changes resulting from the June 2016 storm

Sentinel-2 imagery shows that prior to the storm event the surf zone morphology spanned the typical intermediate states for this beach, from Transverse Bar and Rip to Rhythmic Bar and Beach depending on location alongshore (Figure 3a). It can also be seen that the surf zone was wider along the northern sector relative to the central and southern sectors. As a result of the storm, a surf zone bar ~200 m seaward of the shoreline formed, and is evident in Figure 3b, this is some 50–100 m further seaward than the typical surf zone bar position for this beach. This was most notable in the central and southern parts of the embayment, while in the north surf zone change was less evident (Figure 3). Throughout the remainder of 2016, this bar moved landward at average rates of 1–1.7 m/day and began in places to weld on to the shoreline, although this was likely stochastic and was likely interrupted by other high energy events during remainder of 2016 (Figure 3c, d). By late May of 2017 (~1 year later), the storm bar was no longer distinguishable, and the beach had returned to Rhythmic Bar and Beach or Transverse Bar and Rip at the southern end (Figure 3e). Two bathymetric surveys were captured 2 years before and after the June 2016 storm event, and although they do not overlap the changes observed in the Sentinel-2 images, they demonstrate that upper shoreface morphology is dynamic on annual to interannual timescales down to at least 10 m water depth. These surveys also show that the surf zone is typically wider in the north (SB), and that upper shoreface adjacent to WS had less volume in 2018, following

the storm event, while P4 and SB are more balanced with respect to net volume (Figure 3f).

### June 2016 storm impact on the beach-foredune

The June 2016 storm had a substantial impact along most of the Bengello shoreline. The beach was eroded down vertically and shifted landwards. The vegetated ramp to the foredune was eroded, as was the seaward of the two foredune crests at P1-P4 and WS (Figure 3b, d; Figure 4b). This resulted in the formation of a near vertical sand cliff (scarp) (Figure 2b, d; Figure 4e). The average height of the scarp at profiles P1 to P4 and WS was 2.6 m (2.4–2.8 m) with the base of the scarp also being at a consistent elevation, average 2.3 m (2.1–2.5 m). However, this scarping did not affect the northern 1–2 km of Bengello beach (Figure 4a, d), instead at this location, the storm waves removed the beach berm (−34 m³/m) but deposited sand above 3 m AHD (+16 m³/m) such that the net change across the whole profile was a loss of just 18 m³/m (Figure 2d).

Along the central and southern parts of the embayment, beach-foredune volumes and shoreline positions were substantially reduced. For profiles P1-P4, volume loss averaged 105 m³/m while at WS ~100 m³/m of sand volume was removed. Dividing the volume above and below 3 m AHD, which broadly represents the interface of the beach and foredune at Bengello (McLean & Shen 2006), the volume below 3 m contributed on average 57% of the total volume change at P1-P4, while for WS it represented 69% of the total volume change. Landward shifts of the shoreline at P1-P4 were also substantial with the MSL intercept shifting an average of 36 m inland and the +1 m, +2 m and + 3 m intercepts 37 m, 29 m and 11 m respectively. At WS, the MSL intercept shifted 45 m inland while the +1 m, +2 m and + 3 m intercepts moved inland 42 m, 23 m and 8 m respectively.

### Recovery from the storm

The recovery by volume from the June 2016 storm took nearly 3 years (average of 35 months) with P1-P4 attaining a peak of 60 m³/m above the long-term average in early April–May of 2019. This was still 14 m³/m below the anomalously high pre-storm volume of 74 m³/m (Figure 2e). However, this post-storm recovery volume exceeded the average volume of +27 ± 11 m³/m above the long-term average over the years leading up to the storm (2012–2016) (Figure 2e). Combining field observations of vegetation change with the profile data (Figure 4e) and intercept data (P2 data is shown in Figure 2f as representative of P1-P4), three broad recovery phases can be recognised. Firstly, rapid accretion of the lower beachface took place in the first few months immediately after the storm. By October 2016, the MSL and + 1 m intercepts had returned to close to their pre-storm positions accompanied by an equally rapid increase in lower beach volume (Figure 2e, f). In late 2016 and early 2017, storm events caused beach erosion at Bengello, especially two short-duration storm events in April and May 2017. Again, these events were followed by rapid recovery and the achievement of pre- June 2016 MSL and + 1 m intercept positions (Figure 2f), without any loss of dune features (e.g. Figure 4). This first phase corresponds with the landward migration of the storm bar recorded in Sentinel-2 imagery and suggests that initial beach recovery occurred through onshore transport of sediment eroded from the beach and temporarily stored offshore. This first phase ended in October 2017, by which time the pioneer vegetation, particularly sea rocket, had colonised the first 1–2 m seaward of

the base of scarp and extended laterally along much of the beach This also can been seen in Figure 4, with increases in backshore elevation (or even incipient dune formation in the north, see Figure 4a) for the 2017 TIN surface along the embayment.

The second phase of recovery was from October 2017 to November 2018 and was characterised by vertical build-up of the berm and backshore indicated by the incremental increases in beach volume (Figure 4a-c) and seaward movement of the +2 m and + 3 m intercepts (Figure 2f). In January 2018 a second discontinuous line of sea rocket with *Spinifex* developed 4–5 m seaward of the scarp and later started to expand inland towards the scarp resulting in nebkha on the backshore which still retained its subhorizontal morphology.

The third phase of recovery (November 2018 to May 2019) was characterised by both increased sand volume, development of incipient dunes (in some locations, Figure 4a, b) and seaward progression of all intercepts. This final phase involved the rapid accumulation of a wedge of sand at the back of the beach initially in the zone colonised by sea rocket, including shadow dunes associated with the mounds around each plant. Shortly after, this wedge accreted as an inclined ramp that by April–May 2019 had reached the top of the degraded foredune scarp and through wind deflation was beginning to move sand over the foredune (see photo Figure 2c; LiDAR sequence in Figure 4). Importantly, phases two and three are mostly distinguished according to changes in the upper part of the profiles (above 2 m). The magnitude of beach change represented by the 0 m and + 1 m intercepts greatly exceeds the more subtle accretion of sediment and associated changes in backshore morphology represented by these phases.

Despite the essentially complete recovery by volume, there was a major difference in the geographic distribution of sand across the profiles, that is, beach and foredune recovery was not uniform. Instead, at the central profiles (P1-P4) on average the foredune only recovered ~78% of its pre-storm 2016 volume, whilst the beach accumulated an additional ~12% more sand than its initial volume. This is represented by the change in morphology comparing the pre-erosion survey with the recovered survey for each profile (Figure 2d, Figure 4a-e) where although the foredune recovered with a gently sloping backshore (see photo in Figure 2c), the distinct double crest was not rebuilt. Thus, the foredune by April–May 2019 was far narrower than previously and is particularly evident spatially in the DEMs (cf. 2011 and 2020 in Figure 4b).

## Discussion

### June 2016 storm intra-embayment variability and surf zone-beach interactions

The June 2016 storm event at Bengello Beach, and more broadly along the NSW coast, was a single extreme storm event characterised by an anomalous ENE wave direction. This resulted in uncommon spatial variation in impacts along the embayment. While the northern end of the beach typically experiences the highest exposure to common ESE-SSE storm wave directions, the unusual easterly wave direction of the June 2016 event resulted in wave energy convergence around Broulee Island and an associated reduction in wave energy along the northern end of the beach, as shown in the wave modelling along the 10 m depth contour (Figure 1a). Correspondingly, much less erosion was observed at SB compared to other profiles, with only subtle beach lowering (Figure 2; Figure 4). A wider pre-storm surf zone, as shown in the May 2016 Sentinel imagery (Figure 3) would have dissipated the already reduced incident wave energy farther

from shore, further contributing to lower energy at the shoreline. The lower exposure of the SE facing northern end of the embayment to the storm wave energy and corresponding reduced erosion observed at Bengello Beach, parallels the regional scale patterns presented by Harley et al. (2017) for central and northern NSW. These findings emphasise the sensitivity of beach response to anomalous storm wave directions, particularly where coastline and beach shoreline orientations are oblique to the predominant wave direction, and thus alongshore gradients in beach, dune and surf zone morphology develop under typical conditions (Doyle et al., 2024).

The alongshore variability in morphologic response and morphodynamic feedback between the dunes, beach and surf zone at Bengello Beach before, during, and after the June 2016 event and other storm events (Oliver et al., 2024), raises further questions regarding the role of high energy events in redistributing sand within the embayment over interannual to decadal timescales. The data presented in this study suggests that the northern end of the embayment (SB profile) has been accreting since 2014 while the southern end has been in deficit and retreating over the period 2011–2020 (Figure 3f; Figure 4). It is possible that the transport of large volumes of sand from the beach and dunes to the surf zone in the June 2016 event, and the formation of a deeper storm bar that persisted for months-years following (Figure 3a-e), is one condition that may enable the net transport of sand northward under the typical modal and higher energy ESE-SSE wave conditions.

### Impacts of single extreme storms *versus* successive moderate events

The return to more typical wave conditions in the months-years following the storm allowed for full recovery of the beach-foredune system. This general sequence is illustrated conceptually in Figure 5 where the system moves from (a) – (e). Firstly a wide, well-vegetated foredune and prominent berm is attacked under swash and collision regimes causing beach erosion and foredune scarping (Masselink & van Heteren 2013). Slumping of the foredune scarp and initial beach rebuilding occurs through onshore transport from the surf zone, followed by the transfer of sand to the backshore, eventually forming a vegetated ramp or incipient dune up to the foredune crest. Several alternative pathways are indicated, for instance, moving from (d) back to (b), if a storm of moderate intensity interrupts the recovery process (Figure 5). Another pathway, which was observed and documented in 2022 (Oliver et al. 2024), involves the removal of the foredune under a collision and overwash regime (Masselink & van Heteren 2013) during successive small/ moderate storms. In this scenario (Figure 5di-diii), each successive storm can attack the existing foredune scarp and remove more material with overwash potentially contributing to vegetation dieback and thus reduced effectiveness in sand binding during a later storm (Silva et al., 2016; Maximiliano-Cordova et al., 2019; Davidson et al., 2020; Hesp, 2002).

For the central profiles at Bengello (P1-P4), the single extreme storm event of June 2016 eroded an average sand volume of 105 $m^3$/m, while the 2022 storm sequence removed an average of 78 $m^3$/m. Despite a lower total erosion volume, the 2022 storm sequence resulted in more severe dune impacts, demonstrating that erosion volume alone is not necessarily a reliable indicator of beach-foredune morphological impacts. In both cases, the antecedent (pre-storm) beach-foredune profile had an important influence on erosion impacts, with the fully accreted beach profile in 2016 affording a buffer to moderate dune impacts, while the depleted profile that developed during the 2022 storm sequence ultimately

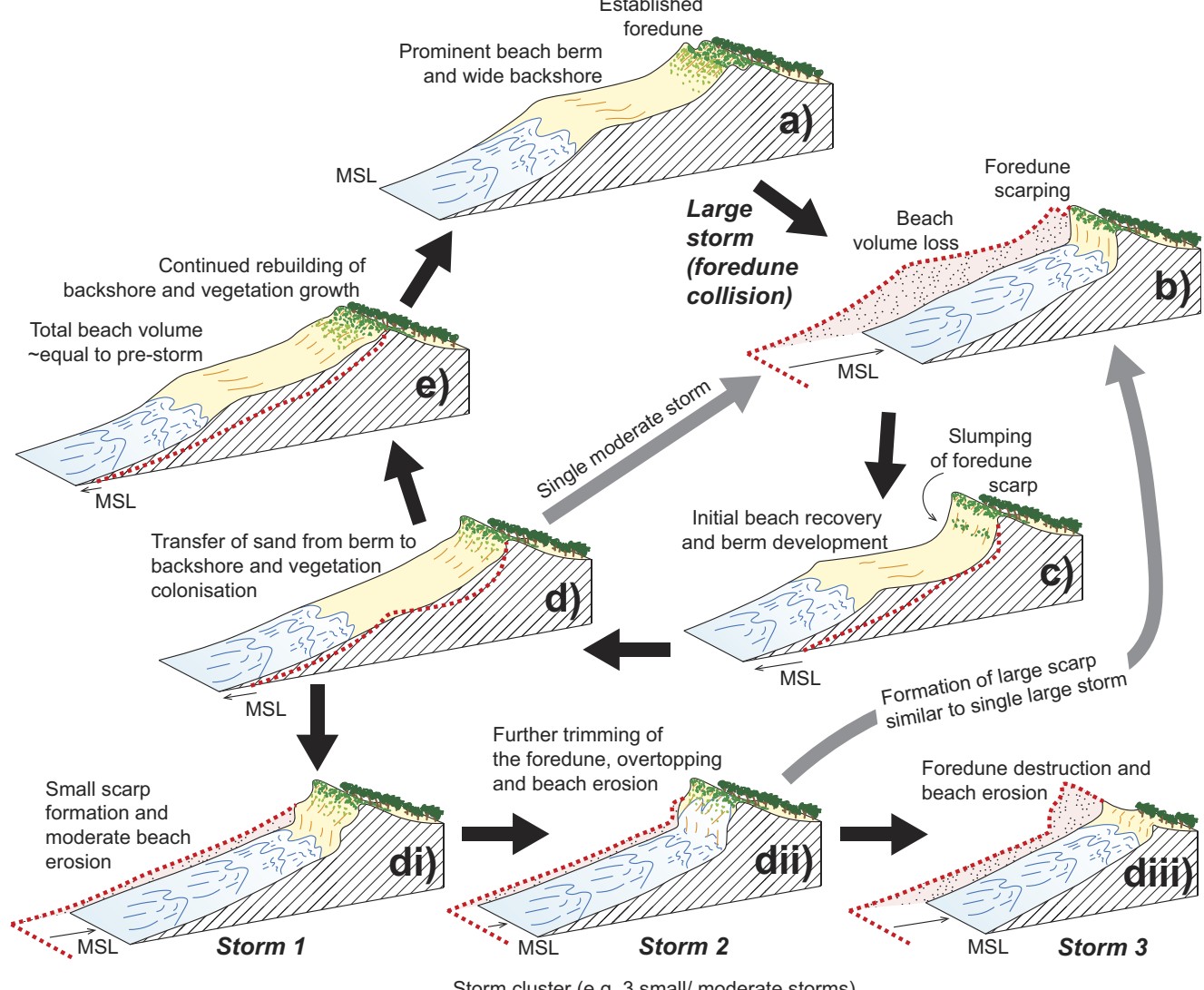

**Figure 5.** Conceptual model characterising erosion impact of single large storm and the subsequent recovery **(a-e)**, and the impact of a sequence of storms (storm cluster) on a partially recovered beach/foredune system **(d-diii)**. The idealised beach/foredune topographic profile from the preceding phase is indicated with the dashed red line. These two situations are generalised versions of what occurred at Bengello during the June 2016 storm and recovery **(a-e)** and in 2022 **(d-diii)** with the erosion, overtopping and destruction of the foredune at some locations as documented previously in this journal (Oliver et al. 2024). Note that there are several other pathways (grey arrows) to the highly-eroded profile shown in **(b)**, one from a single moderate storm occurring when the system is in phase **(d)**, and another from the storm cluster where **(dii)** leads to **(b)** rather than **(diii)**.

allowed more extensive wave attack, scarping and even destruction of the foredune (see Figure 3, Oliver et al., 2024) by comparatively moderate storm conditions (as shown elsewhere in NSW by Splinter et al., 2018 and Beuzen et al., 2019). The comparative behaviours and impacts are illustrated in (Figure 5) which shows how foredune impacts may vary depending on the sequencing of storms over months to years, and particularly, the opportunity (or lack thereof) for beach recovery processes to develop an accreted beach-foredune profile between storms. An accreted beach offers an important sacrificial sand volume that can be rapidly mobilised during the onset of storm conditions and formed into surf zone bars that moderates the storm wave energy reaching the shore.

While there is no consistent definition of what constitutes a storm cluster (Eichentopf et al. 2019), analysis by Eichentopf et al. (2020) found that storm sequences do not necessarily cause greater beach erosion than a single event and suggest that pre-storm beach-foredune morphology is probably most critical in determining

storm impacts. For Narrabeen-Collaroy, Karunarathna et al., (2014) found that storm clusters of two or three storms caused beach erosion comparable to 1 in 10 year and 1 in 48 year return periods respectively. At Bengello, the single extreme June 2016 event encountered a fully accreted beach-foredune profile that offered high resilience to the extreme nearshore wave conditions sustained over multiple days. In contrast, the fifth (July) storm in the 2022 sequence encountered a depleted beach profile with reduced resilience and a scarped foredune left by preceding storms, particularly the largest storm of the sequence only 5 weeks earlier (Oliver et al. 2024). Comparison between the storm characteristics and responses of Bengello Beach to the June 2016 storm and 2022 storm sequence (Oliver et al., 2024) highlights that detailed analysis of storm wave and water level conditions, and even the sequencing of storm events, may be nevertheless insufficient to explain (and predict) storm impacts on beaches without an adequate understanding of the antecedent beach-foredune condition in each case.

### *Beach-foredune recovery and future resilience of sandy coasts*

The results of this study highlight the protective role that wide beaches, and natural, vegetated foredunes play in buffering against the impact of extreme storms. During June 2016 storm at Bengello, the wide beach was an initial buffer to the foredune, but once this buffer was diminished and waves reached the foredune, then the foredune itself became a buffer to the land behind. Importantly, had the June 2016 storm impact not been absorbed by the wide beach, and then in turn by the backshore and well-vegetated foredune, the beach-foredune morphology and volume would have resembled close to that which occurred following the notable storms of 1974, 1976 and 1978 which still 'stand out' in the >50-year survey record (McLean et al., 2023).

While Bengello Beach recovered by volume following 2016, the foredune was narrower than before, fundamentally altering the resilience of central and southern parts of the beach. This modification to the foredune became important when storms occurred in 2020, 2021, so that by 2022, the persistent erosion of the foredune by multiple storm events without sufficient recovery, led to partial or complete foredune destruction by only a moderately energetic event within a sequence (Oliver et al. 2024). In this sense, the foredune destruction in 2022 for the central profiles (P1-P4) was in part the result of morphological changes to the system which can be traced back at least to June 2016 storm. In terms of SB and WS beach-foredune volumes, WS had experienced a substantial erosion event in mid-2014 (Figure 2e), and although it had just recovered in volume by the time of the June 2016 event, by 2020, beach volume in the south (WS) remained lower than in April 2016. Conversely, SB appears to have been steadily gaining sand since mid-2014 such that by the beginning of 2020 it was 82 $m^3$/m greater than the pre-2016 storm volume (Figure 2e).

Beaches and vegetated foredunes therefore function as dynamic sediment reservoirs that together buffer coastal assets by moderating wave impact. This study reinforces the importance of antecedent beach, foredune and dune plant conditions and their role in enhancing coastal resilience to extreme events (Hesp, 2002; Psuty, 2008; Zarnetske et al., 2015; Feagin et al., 2015; Davidson et al., 2020). Where successive storms compress recovery windows, lower beach volumes may lead to wave attack of foredunes which do not sufficiently recover. As a result, they may become narrower and lower, reducing their protective capacity and increasing coastal vulnerability until rebuilding via sand transport from a wide beach and revegetation can occur (Keijsers et al., 2016; Davidson et al., 2020). Therefore, management priorities that maintain or enhance natural beach and dune width and/ or volumes, and connectivity to the wider coastal area are critical for the future as coasts globally respond to a range of pressures. These may include, facilitating beach-foredune sand exchange, avoiding "coastal squeeze" (Lansu et al., 2024), controlling trampling and access, stabilising foredunes with native vegetation, and applying soft engineering such as dune nourishment and nature-based "living shoreline" measures. (Temmerman et al., 2013; Doyle and Woodroffe, 2023; Davidson-Arnott et al, 2024; Morris et al., 2024).

## Conclusions

The June 2016 ECL caused widespread beach erosion along the NSW coast including at Bengello Beach – the site of a multidecadal beach-foredune monitoring program. The impact of this event has been quantified with traditional beach survey methods at six separate profiles along the beach and supplemented with airborne LiDAR capturing the pre-storm beach-foredune morphology, immediate post-storm impact, and recovery phases. This data shows that, except for the northern profile (SB), ~100–110 $m^3$/m of sand was eroded from the beach and foredune during this single storm event and a wide, double-crested and well-vegetated foredune was cut back forming a 2.6 m scarp. Modelled nearshore wave conditions explain the alongshore variability of the storm impact, with the northern end of the embayment partly sheltered from the unusual ENE storm wave direction. In response to this storm event, Sentinel2 imagery shows an elongate subaqueous surf zone bar formed 50–100 m further seaward than typical surf zone bar positions and gradually moved landwards during post-storm recovery, facilitating beach and then dune rebuilding through landward sediment exchange. Recovery from the storm took ~35 months for all profiles except SB. However, the double-crested foredune was not rebuilt and the overall foredune became narrower. Overall, the impact of this event, and comparison with later events in 2022, demonstrate that a wide beach and stable, well-vegetated foredune are critical buffers against extreme single storms. Conversely, a narrow beach and narrow or low foredune is vulnerable to substantial erosion and even destruction with even moderate storm events as was observed at some locations at Bengello in 2022. These results highlight the need for coastal managers to invest in maintaining wide beaches with well-vegetated foredunes with sufficient combined volume to absorb the impact of extreme single storms and storm clusters.

**Open peer review.** To view the open peer review materials for this article, please visit http://doi.org/10.1017/cft.2025.10015.

**Data availability statement.** Deepwater wave data are available on request to Manly Hydraulics Laboratory (MHL), or from the Australian National Wave Archive (IMOS, 2023). The 2018 Marine LiDAR is available from https://datasets.seed.nsw.gov.au/dataset/marine-lidar-topo-bathy-2018. The 2011 NSW Spatial Services terrestrial LiDAR is available from https://elevation.fsdf.org.au/.

**Acknowledgements.** The authors thank Fugro Australia Pty. Ltd. for the 2018 marine LiDAR data acquisition and processing, NSW Spatial Services (NSW Government) for commissioning and processing the 2011 LiDAR, as well as Jason Middleton and Peter Mumford, of UNSW Airborne Lidar Facility, School of Aviation, UNSW Australia for data acquisition and processing of other LiDAR surveys. We thank the NSW DCCEEW for commissioning, processing and providing the LiDAR data collected by Fugro and UNSW. Offshore wave buoy data at Sydney and Eden were collected by Manly Hydraulics Laboratory on behalf of NSW DCCEEW.

**Author contribution.** **Thomas Oliver:** Writing, reviewing, editing, figure conceptualisation and drafting, data analysis (field survey data, Sentinel-2). **Michael Kinsela:** writing, reviewing editing, figure conceptualisation and drafting, data analysis (wave modelling, wave data). **Thomas Doyle:** Writing, reviewing, editing, figure conceptualisation and drafting, data analysis (LiDAR DEMs). **Dylan McLaughlin:** data analysis (LiDAR DEMs, DEMs of difference). **Roger McLean:** writing, reviewing, editing, data collection (original survey data, photos), data analysis (field survey data).

**Financial support.** This research received no specific grant from any funding agency, commercial or not-for-profit sectors. The School of Science at UNSW Canberra supported the field components of this research.

**Competing interests.** The authors declare none.

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
