## [Reviewer Report]

Thank you for the opportunity to review this valuable manuscript. This is a fascinating study that leverages long-term survey data and modern remote sensing techniques to elucidate the impacts of the extreme 2016 storm event at Bengello Beach and the detailed recovery process thereafter. The demonstration of the spatially heterogeneous response of the beach-dune system to a storm with an anomalous wave direction is a significant contribution to coastal research.

The manuscript is well-written overall, but I believe a few revisions could further enhance its impact. My general and specific comments are provided below for your consideration.

General Comments

The most critical finding of this paper, as I interpret it, is the demonstration of the crucial relationship where the pre-storm beach volume determines the degree of erosion the foredune sustains relative to the storm’s magnitude.

The case of the 2016 extreme storm is particularly insightful. At that time, despite the presence of one of the widest beaches in the observational record, significant erosion still occurred, resulting in a scarp averaging 2.6 m in height on the foredune. This fact indicates that the foredune did not simply withstand the storm. Rather, a more accurate interpretation would be that the vast beach acted as a “sacrificial buffer,” absorbing the majority of the wave energy, thereby preventing a more catastrophic outcome, such as the complete destruction of the foredune.

This situation provides a stark contrast to the conditions in 2022, when the beach volume was insufficient during the recovery phase. In 2022, even a moderate storm easily caused erosion because the depleted beach could not serve as an effective buffer, allowing wave energy to impact the foredune directly.

This comparison—2016, where a sufficient buffer led to suppressed damage even during an extreme storm, versus 2022, where an insufficient buffer resulted in significant damage from only a moderate storm—is the core value of this research. The current abstract and conclusion seem to overstate the protective function of the foredune itself. Re-framing the narrative from a more mechanistic viewpoint—that the resilience of the foredune is highly dependent on the dynamic depositional state of the beach in front of it—would further clarify the novelty and impact of this study.

Major Comments

On the Definition of the Three Recovery Phases (P.8, L.41)

The text states that three broad recovery phases are evident from the P1-P4 profile data. However, it is difficult to clearly distinguish these three phases from the time-series of shoreline intercepts presented in Figure 2f. The +0m and +1m intercepts, in particular, are highly variable, and the rapid recovery noted as Phase 1 appears to be within the range of natural variability observed at other times. Conversely, the +2m and +3m intercepts seem to show a consistent, gradual recovery post-2016, without clear steps corresponding to the phases. While the recovery process described by the authors is likely valid based on LiDAR and qualitative observations, the evidence from the intercept data alone is weak. It is necessary to more clearly describe the criteria used to define these phases (e.g., based on specific morphological changes, rates of volume change from LiDAR, etc.) to provide an objective basis for the division.

On the Interpretation of Citations and Deepening the Discussion (P.10, L.33)

The paper cites Eichentopf et al. (2019, 2020) to support the “importance of a wide ‘healthy’ pre-event beach/foredune system.” As you note, these studies suggest that a larger pre-storm volume can lead to greater erosion potential, and that storm clustering can limit recovery opportunities. This context actually reinforces the mechanism proposed in this paper: that a large beach volume acts as a buffer to protect the foredune. The current description feels somewhat oversimplified. A more precise reflection of the cited works and a deeper discussion from the perspective of the “buffer capacity” of the pre-storm beach volume would create a more compelling argument.

Specific Comments

Clarity and Consistency of Terminology:

・ P.5, L.24/L.45: If “Marine LiDAR” and “seamless topographic-bathymetric survey” refer to the same 2018 dataset, the terms should be unified or their relationship clarified upon first use to avoid reader confusion (e.g., “A seamless topographic-bathymetric LiDAR survey collected in 2018...”).

・ P.2, L.44: The term “multidecadal” could be misleading when applied to high-resolution datasets like LiDAR. It would be clearer to distinguish the timescales of different data types, e.g., “By combining multidecadal field survey data with recently accumulating high-resolution airborne LiDAR datasets...”

Readability with Figure Citations:

・ P.3, L.49: When describing the dominant SSE wave direction, citing Figure 1d, which illustrates this, would be helpful.

・ Methodology Section: Readability would be greatly improved by citing relevant figures within the methods descriptions.

Storm wave modelling (P.4) → Fig 1b, 1c

LiDAR collection (P.5) → Fig 4

Beach surveys (P.6) → Fig 2d, 2e, 2f

Accuracy of Descriptions:

・ P.3, L.54: Since data from the Sofar buoy is not used in this study, this sentence should be removed or its relevance justified (e.g., for context on future work).

・ P.5, L.40: It should be explicitly stated in the Methods section that the 2016 LiDAR survey was conducted after the storm.

・ P.7, L.31: Please clarify if “9.08 MWhr/m” is an average value along the 30 m contour or a value at a specific point.

・ P.7, L.48: As Figure 3b is from July 19th (over a month after the storm), the phrase “During the storm” is inaccurate. “Following the storm” or “As a result of the storm” would be more appropriate.

・ P.8, L.38: The statement that the recovery volume “exceeded the average volume” in 2012-2016 would be more precise if stated as exceeding the “average accretion volume (i.e., the deviation from the long-term mean)” for that period.

Scope of Conclusions:

・ P.11, L.19: This study does not directly demonstrate the influence of the dune and its vegetation on beach recovery. Rather, it shows how beach recovery contributes to dune recovery. The statement should be revised to accurately reflect the scope of the findings to avoid misinterpretation.

・ P.11, L.54: I strongly recommend revising the statement to reflect that the large volume of the pre-storm beach provides the critical buffering function that protects the foredune, rather than attributing the buffering role to the foredune itself. This is a more mechanistic and accurate conclusion.

I hope these comments are helpful in strengthening this excellent manuscript.

---

## [Reviewer Report]

The impact of storms on beaches is an important factor in understanding the dynamic geomorphological evolution of sandy coasts. Therefore, this article adopts multiple research methods, multiple observations, and long-term data collection to obtain relevant information, attempting to clarify the impact of a huge storm in 2016 on beaches, especially the research work on its subsequent three recovery stages, which has certain significance.

Overall, the article has a clear and logical discussion, sufficient data, clear and beautiful graphics, and has a foundation for publication.

The following questions are for reference:

1.Introduction

The introduction section provides a brief overview of existing research and does not fully explain the unique contribution of current research in the field of the impact of extreme storms on coastlines. For example, the literature mentions the importance of vegetation on the front sand dunes for storm buffering, but lacks a specific analysis of the limitations of existing research, such as the controversy over the relationship between vegetation cover and sand dune stability.

2.Methodology

It is unclear the location of 6 of the 7 permanent offshore wave buoys.

In Figure 3f, it said that the bathymetric profiles (locations shown in e) are derived from two data sources. The 2014 data is a single-beam beach boat and jet ski survey, and the 2018 data is from marine LiDAR. How to combine two data sets? How to connect the data above -2 m and below -2 m?

The LiDAR data processing flow does not clearly specify how to classify ground points (such as “bare ground”) and the validation method. For example, the article mentions “using Terra Solid tool for classification”, but does not explain the classification criteria or error range.

3. Results and Discussion

In this section, the article elaborates on the process of the impact of the storm on the beach, but this still belongs to the normal influence process of coastal dynamic geomorphology on the storm surge of the beach, such as the formation of underwater sandbars on the outside of the beach, and then the sandbar helped the beach to be recovered through wave motion.

However, the article discusses the erosion and restoration process of the entire hydrodynamic-beach-sand dune system, especially the specific restoration time stages, which is a very meaningful and detailed process record.

One characteristic of the article is its focus on the impact of sand dune vegetation coverage, which plays a very important role in the restoration of beaches and sand dunes. However, this is only a direct observation based on satellite images and has not been associated with long-term monitoring data. Can existing monitoring data reflect this relationship?

---

## [Reviewer Report]

Lines 14-15: RE: “Existing field-based long-term monitoring programs are sparse, both globally and regionally, and yet have still provided critical data on storm impacts and recovery in the context of decadal trends (e.g. Banno et al. 2020;”

This list fails to include other very long term monitoring sites and data including:

Hesp, P. A. (1989). A review of biological and geomorphological processes involved in the initiation and development of incipient foredunes. Proceedings of the Royal Society of Edinburgh, Section B: Biological Sciences, 96, 181-201.

Hesp, P. A. (2013). A 34 year record of foredune evolution, Dark Point, NSW, Australia. Journal of coastal research, (65), 1295-1300.

Study site Line 30: :ADD in caps: “Inland of Bengello Beach is a ~2 km WIDE sequence”

P4 line 12: RE: “The coincidence of large tides with a moderate storm

Surge” State what the storm surge height was.

P6 lines 22-23: Where not were

P6 line 46 Remove “Healthy” – totally subjective word

Line 52: be consistent!! Change frontal dune to foredune and make sure this is done elsewhere in the ms if necessary

P8 line 59: ADD in caps: “The second phase of recovery was from October 2017 to November 2018 and was characterised BY…

P9 line 6: RE: “started to expand inland towards the scarp resulting in patchy mounds on the backshore” Replace “patcjy mounds” with ‘nebkha’

RE: At Bengello, had the June 2016 storm impact occurred under average beach volumes, there would have been a return to conditions close to those which occurred following the notable storms of 1974, 1976 and 1978 which still ‘standout’ in the >50-year survey record (McLean et al., 2023).”

Its not clear to me what u mean here. It took a significant no of years for beaches to recover from the 1974 and 78 storms and far longer than it took for this 2016 storm so what do u mean? Rewrite.

cheers

patrick

---

## [Editor Report]

We have received 3 good quality reviews of your paper, and all of the reviewers recommend minor revision. Their comments are mainly to add further clarity and some editorial changes, and I do not foresee any difficulties in your ability to address the comments and submit a revised paper relatively quickly.

---

## [Editor Report]

I have read through all your responses, and I am happy that the reviewers concerns have been adequately addressed. I look forward to seeing your paper published.